# Residual Stress Engineering for Wire Drawing of Austenitic Stainless Steel X5CrNi18-10 by Variation in Die Geometries—Effect of Drawing Speed and Process Temperature

**DOI:** 10.3390/ma17051174

**Published:** 2024-03-02

**Authors:** René Selbmann, Jens Gibmeier, Nicola Simon, Verena Kräusel, Markus Bergmann

**Affiliations:** 1Fraunhofer Institute for Machine Tools and Forming Technology (IWU), D-09126 Chemnitz, Germany; verena.kraeusel@iwu.fraunhofer.de (V.K.); markus.bergmann@iwu.fraunhofer.de (M.B.); 2Institute for Applied Materials (IAM-WK)—Materials Science and Engineering, D-76131 Karlsruhe, Germanynicola.simon@kit.edu (N.S.)

**Keywords:** wire drawing, drawing speed, residual stress modification, FE simulation, residual stress analysis

## Abstract

As a result of conventional wire-forming processes, the residual stress distribution in wires is frequently unfavorable for subsequent forming processes such as bending operations. High tensile residual stresses typically occur in the near-surface region of the wires and can limit further application and processability of the semi-finished products. This paper presents an approach for tailoring the residual stress distribution by modifying the forming process, especially with regard to the die geometry and the influence of the drawing velocity as well as the wire temperature. The aim is to mitigate the near-surface tensile residual stresses induced by the drawing process. Preliminary studies have shown that modifications in the forming zone of the dies have a significant impact on the plastic strain and deformation direction, and the approach can be applied to effectively reduce the process-induced near-surface residual stress distributions without affecting the diameter of the product geometry. In this first approach, the process variant using three different drawing die geometries was established for the metastable austenitic stainless steel X5CrNi18-10 (1.4301) using slow (20 mm/s) and fast (2000 mm/s) drawing velocities. The residual stress depth distributions were determined by means of incremental hole drilling. Complementary X-ray stress analysis was carried out to analyze the phase-specific residual stresses since strain-induced martensitic transformations occurred close to the surface as a consequence of the shear deformation and the frictional loading. This paper describes the setup of the drawing tools as well as the results of the experimental tests.

## 1. Introduction

The most important manufacturing processes for elongated, bar-shaped components with a small cross-section are wire drawing and impact extrusion. The main difference between these processes lies in the position and direction of force application. Residual stresses occur in the material due to elastic–plastic deformation, which is inhomogeneously distributed over the cross-section after removing all external forces. These residual stresses can affect subsequent forming operations of the semi-finished products, for instance, bending operations and the mechanical load-bearing capacity of the wire products [1]. Figure 1a shows the schematic setup of the wire drawing process. The arrow pointing in the axial direction indicates the movement direction of the material with the drawing speed and the concurrent impact of the drawing force. The geometry of the contour is cone-shaped, which is primarily determined by the taper angle (2α). The forming zone is located in the inlet cone region of the drawing die. According to the cone angle, a triangle of forces results from the acting process force, the shear force, and the normal force. The resulting vector between the normal force and the lateral force additionally depends on friction. The forces cause radial, axial, and tangential stresses in the material. According to the force triangle, the deformation is predominantly affected by the normal force and by resulting radial and tangential compressive stresses generated during processing and, to a smaller extent, by the external drawing force [2]. As a result of the elasto-plastic deformation during the wire drawing process, tensile residual stresses are generated near the surface in the axial and tangential direction, and balancing compressive residual stresses form in the core (Figure 1b). These characteristic residual stress distributions are unfavorable for subsequent forming operations, such as bending of torsion bar springs. In the production process for torsion bar springs, these residual stresses specifically limit the formability, e.g., in bending operations with small bending radii. Here, the tensile residual stresses close to the surface add up to the tensile load stresses of the bending deformation according to the superposition principle and can cause early failure, e.g., due to the formation of cracks in the highly stressed locations. Furthermore, even in cases where no early cracking occurs right after processing, these residual stress distributions can reduce the fatigue strength and fatigue life of the components [3]. By mitigating the process-induced residual tensile stresses in the near-surface region of the wires while drawing, smaller bending radii can be achieved, and the load-bearing capacity can be increased. Ideally, compressive residual stresses should be preferred in order to improve the fatigue strength since they normally contribute to a delayed or suppressed crack growth. If the residual stress distribution is adapted to the influences of the external forces or torque stress distribution, an optimum result can be achieved.

In order to counteract residual stress distributions with high tensile residual stresses in the near-surface region, typically, a costly heat treatment is applied between the forming stages. Alternatively, material-dependent reshaping or a second forming stage with less than 0.8% relative reduction in the cross-section [6] is performed, and/or mechanical post-treatments are applied as, e.g., shot peening. In any case, these measures serve to minimize or mitigate the tensile residual stresses near the surface, as these can lead to premature damage during further processing of semi-finished products. In order to prove the effectiveness of these measures, it is necessary to analyze the process-related depth distributions of the residual stresses using suitable techniques. In case multiple phases show up in the microstructure, phase-selective methods must be chosen. In the present case for the metastable austenitic steel X5CrNi18-10 (1.4301), deformation-induced martensitic transformation might occur. This should lead to a microstructure that consists of two phases: austenite and martensite. In this case, only diffraction methods such as X-ray residual stress analysis are appropriate to account for the two-phase structure. Hence, in order to evaluate the manufacturing strategy properly, it is necessary to perform phase-specific residual stress analyses in the affected region, presumably occurring only in the very-near-surface region in all phases involved. Meaningful phase-specific residual stress analysis using lab-based methods can be expected if the volume content is at least 10%. 

It is well known that the geometry of the tool has a great influence on the resulting residual stress distribution. Hence, adapting tool geometry represents a suitable adjustment parameter for tailoring local residual stress distributions. Tekkaya [6] described and compared how residual stresses were influenced by changing the percentage reduction in the wire diameter in one conical die. Celentano et al. [7] demonstrated that by minimizing the drawing steps, a decrease in the residual stresses could achieved. Överstam [8] investigated the influence of manufacturing tolerances and die geometry on the residual stresses, however, without using special forming elements. Atienza et al. [9] and Siva et al. [10] determined the resulting residual stresses, which arise by modification of the taper angle and form of the die compared.

A further approach for influencing residual stress distribution during the forming process is applying methods of gradation extrusion and gradation rolling. These were developed to create tailored properties in metallic materials and can now be adapted to wire drawing [11,12]. Gradation extrusion promises to be particularly suitable, with additional severe plastic deformation (spd) elements being integrated into the die. These elements create a material flow with multiple local direction changes along the surface contour of the tool. The forming process results in certain property modifications with gradients in microstructure and mechanical characteristics across the workpiece cross-section with property changes localized in the surface area of the component [13,14]. Preliminary work on austenitic steel 1.4301, which is a typical material for technical springs, is progressing. Technical springs are subject to complex load conditions during operation. For the basic investigation, the focus was first only on the rod-shaped cylindrical part. Flow curves and the necessary approximations for higher degrees of plastic strain were developed from experimental tests and presented in [15].

Baumann et al. [16] developed various tool geometries with specific shaped elements for wire drawing and examined them on the basis of finite element (FE) studies. The aim of the investigation of conventional, convex, and concave forming elements was to minimize the tensile residual stresses in the near-surface region of the wires. Selbmann et al. [17] investigated these selected geometries experimentally and by additional FE simulations for an extrusion process. The results indicated that the process-induced tensile residual stresses close to the surface could be significantly reduced using the specific die geometries. According to Bauman et al. [18], the die geometries were used for the design of wire drawing tools and for the application of the wire drawing process, as presented in Figure 2. 

The geometric elements implemented in the forming zone cause a high degree of plastic deformation during the forming process, which also changes the residual stress distribution along the cross-sectional cut of the parts. However, the final diameter is identical to conventional forming processing. The main geometric dimensions of the tools are identical. A diameter reduction in the specimen from Ø 12 mm to Ø 10.8 mm was considered. The variant of the conventional die geometry represents the reference process without additional spd-elements. Convex geometry includes one convex forming element and the concave geometry a concave forming element (see Figure 2). The application of those specific elements results in a modified material flow. In order to investigate the influence of the geometry elements on the residual stress formation during wire drawing, tools have been developed in which the drawing dies can be inserted. The investigations in this paper were performed with the presented drawing dies without any geometric adjustments. 

Using convex and concave geometry, it could be shown by means of FE simulations and experimental investigations that the tensile residual stresses close to the surface could be influenced and specifically reduced compared to the application of a conventional die geometry [5,18]. In these processes, however, the processing speed of 20 mm/s was atypical for wire drawing, and the effect of near-surface process-related wire temperatures was not considered. Furthermore, until now, only residual stresses at the very surface have been investigated experimentally. However, for a proper assessment of the process-induced residual stresses, data from the surface alone are not sufficient.

In the work presented in this paper, the drawing speed is now increased significantly in order to more closely resemble a series process and to compare the results with a slower process variant. Moreover, residual stress analysis was not only limited to the very surface but also residual stress depth gradients were determined. Pre-studies have shown that for the given geometry, incremental hole drilling is a suitable approach to analyze the macro residual stress depth distribution. Since steel 1.4301 is a metastable austenitic steel, the forming process, with its high plastic strain induced in the near-surface region, causes the deformation-induced transformation of austenite to martensite [19]. To account for this, complementary X-ray stress analyses were performed. Afterward, the phase-specific residual stresses were determined in the near-surface region.

By effectively influencing the residual stress distributions directly during the formation of the semi-finished products, further implications of follow-up processes are possible since post-treatments are not required, e.g., shot peening to adjust the residual stresses and overall, as the processing route becomes significantly more eco-efficient. From the application side, reduced tensile residual stresses in the near-surface layer of drawn wire allow for, e.g., smaller bending radii or an increased resistibility against cracks compared to conventionally manufactured products without any additional process steps to adapt the residual stress distribution. Lower tensile residual stresses at the near-surface layer of the wire in combination with the tensile load stresses during bending, taking into account the superposition principle, smaller bending radii can be realized without the risk of early cracking.

## 2. Materials and Methods

For the experimental tests at a temperature of 20 °C (room temperature), cylindrical specimens made of austenitic steel 1.4301 are used. The chemical composition of the metastable austenitic stainless steel 1.4301 is given in Table 1.

The experimental tests were carried out using these samples, which were annealed at 1050 °C for 15 min and then cooled in still air prior to the drawing experiments. The applied heat treatment serves to achieve uniform initial conditions before forming (Yield strength = 190 MPa). The cylindrical specimens consist of an examined section (diameter = 12 mm, height = 100 mm), which is formed in the drawing experiments, and a part for the assembly to the drawing die and clamping in the tool (diameter = 9.6 mm, height = 225 mm). The transition section between these parts is provided with an angle of 6° to generate a suitable inlet into the drawing die, similar to sharpening in industrial processes. 

The tool setup of the experimental device for the drawing process includes the interchangeable specific die with additional convex or concave geometric elements presented in the introduction, and for further details, it is referred to [5,16,17,18]. A solid lubricant (LOCTITE LB 8191) was applied to each of the samples, and additionally, lubrication with high-alloyed drawing oil was used during forming. For the current investigations, the drawing tools [5,18] were upgraded to include temperature measurements and were further adapted for drawing at higher speeds. In Figure 3 and Figure 4, the drawing tool setups are presented. The experimental investigations presented in this paper are divided into two different drawing speeds to investigate how the wire drawing speed influences the formation of residual stresses on the wire surface while using the same drawing die geometries as in [5,17,18]. Overall, 20 mm/s and 2000 mm/s were selected. For the drawing speed of 20 mm/s, the influence of the wire temperature was investigated. Therefore, the samples were heated in a furnace up to about 200 °C, adjusted in the drawing tool, and drawn as in the other processes. The aim was to realize a temperature during the drawing of about 180 °C, which corresponds to a characteristic temperature for wire drawing. To monitor this temperature during wire drawing, a thermographic camera was used (see Figure 3). The drawing speed of 20 mm/s corresponded to the forming speed of the extrusion tests, which made it possible to compare the results of the extrusion [17] and the drawing process [18], 2000 mm/s corresponds to a typical wire drawing speed for this diameter range. The drawing tool is fixed in a mechanical press, and the return stroke is used to achieve a constant drawing speed.

However, it is known that the classical wire drawing process runs at significantly higher speeds. To account for this, the drawing tool was adapted to a high-speed tensile testing machine (Figure 4) to achieve a drawing speed of 2000 mm/s. In both cases, the specimens were clamped in a specially adjusted clamping unit. The process forces and drawing distance were measured by load cells and position sensors, respectively.

The depth distribution of the macro residual stresses was analyzed by means of incremental hole drilling using a drilling device of type RS200, Vishay measurements group. For drilling, a TiN-coated end mill with a nominal diameter of ø 0.8 mm was applied. The strain relaxations during stepwise drilling of the blind hole were recorded by strain gage rosettes of type EA-06-031RE-120, which were interconnected in a Wheatstone half-bridge with temperature compensation. A carrier frequency amplifier of type PICAS 4K from Peekel Instruments was applied in this regard. Stress calculation was carried out using the differential approach [21]. The measuring points were situated in the middle of the drawn sample, i.e., in the region that was subjected to continuous deformation. These residual stress analyses were supplemented by X-ray stress analyses according to the sin^2^ψ-method [22] on the identical sample at another location on the circumference, i.e., on the opposite side. The phase-specific residual stresses were analyzed for martensite and for the austenite phase for the α′{211}- and the γ{220}-lattice planes, respectively, using V-filtered Cr Kα radiation. On the primary side, a pinhole collimator with a nominal diameter of ø 1 mm was applied. On the secondary side, a 4 mm symmetrizing slit was used [23] for the martensite diffraction lines, and a 2 mm slit in the case of the austenite line. For each measurement point and for each phase-specific residual stress analysis, at minimum, 15 sample tilts in the range between −60° ≤ ψ ≤ 60° equidistantly distributed in sin^2^ψ were considered. The diffraction elastic constants (DEC) for each respective phase given in Table 2 were calculated from the single crystal elastic constants of the stiffness tensor according to [24,25] using the Eshelby–Kröner model [26]. 

For evaluation of the residual stresses by means of X-ray diffraction the X-ray diffraction lines were fitted by Person VII functions after background correction. To calculate the error bars, the error of this peak fitting and the error from determining the slope of the line of the distribution of peak positions vs. sin^2^ψ (linear regression analysis) is used, taking error propagation into account. For the analysis of depth distribution, a successive electrochemical layer removal was carried out in combination with repeated XRD analysis on the newly formed surface. The macro residual stresses and the phase-specific micro residual stresses can be separated from the phase-specific residual stresses using a simple rule of mixture, taking into account the local phase fractions. Hence, the required phase fraction of martensite and austenite was also determined for each electro-polishing step by means of XRD using Nb-filtered MoK*α*-radiation according to the Rietveld method considering a 2θ-range from 20° to 58°. The combination of XRD and incremental hole drilling for residual analysis is advantageous for a meaningful assessment of the residual stress depth distribution resulting from wire drawing.

## 3. Results and Discussion

The experimental investigations were divided into two categories. The temperature influence was examined at room temperature and at 180 °C with relatively low drawing speeds of 20 mm/s. In the following, the drawing speed was increased up to 2000 mm/s at room temperature. The process data of the drawing force and path were recorded, and die geometries designated as ‘conventional’, ‘convex’, and ‘concave’ were used.

### 3.1. Drawing Speed of 20 mm/s—Material at Room Temperature and at 180 °C

The evaluation of the required drawing forces at room temperature shows that a significantly higher drawing force is required with the convex geometry variant due to the high degree of deformation caused by the geometric element, as seen in Figure 5. The peak force, particularly clear with the convex geometry at the beginning of the forming process, can certainly be attributed to the special conditions of the convex geometry. The special element literally protrudes into the material. Since this running-in process only plays a minor role in wire drawing, no explicit investigations were carried out. More likely, the continuous pulling force should be shown, which begins shortly after the start. The conventional and concave geometries require similar forces, which are about 25–30% lower than for the convex geometric elements. At a wire temperature of 180 °C, the force demand for the conventional and concave geometry increases due to the thermal expansion of the wire and because no lubricant could be used at this elevated temperature. For the convex elements, the force demand is slightly decreased compared to the room temperature application. In general, for a speed of 20 mm/s and a temperature of 180 °C, the force-process-time characteristics of the different geometric elements are similar, with a mean force demand of approx. 25 kN. 

The amount of martensite determined close to the surface appears somewhat unexpected at first sight since the higher temperatures were chosen precisely to prevent the martensitic transformation (Figure 6). However, what counteracts this is the fact that at the higher temperature, the lubricant could not be used in the same way as in the case of the tests at room temperature since the lubricant degenerates at the higher temperature of 180 °C.

As a result, higher frictional forces inevitably occur on the immediate surface, which results in a somewhat higher transformation into martensite up to a phase fraction of about 18% for the reference tool geometry. Here, the concave and convex tool geometry results in a slightly lower amount of transformed martensite. However, it must be clearly stated that this martensitic transformation is only limited to the first 10 μm from the surface due to the very localized high degrees of shear deformation. 

In Figure 7, the depth distribution of the macro residual stresses in the axial and tangential direction is presented for the application of the different drawing die geometries for a drawing speed of 20 mm/s at (a) room temperature and at (b) an elevated temperature of 180 °C. It can be clearly seen that significantly higher tensile residual stresses were determined for the macro-residual stresses in the room temperature process variant. An additional XRD stress analysis in the surface area confirms this trend. However, at higher depths the macro residual stresses in all cases stabilize at a rather high tensile level. Here, the highest tensile residual stresses were determined for the convex drawing die geometry, which corresponds well with much higher force demand for this variant (cf. Figure 5a) and the related higher degree of deformation. Even the conventional tool geometry resulted in lower tensile residual stress values. For the convex and the conventional die geometry variant, a similar depth distribution was determined, which again nicely corresponds with similar force demands. However, at least up to the depth of about 150–200 µm, the convex geometry variant resulted in the lowest tensile residual stresses for processing at room temperature at the low drawing speed of 20 mm/s.

The situation changes completely for processing at 180 °C. For all tool geometry variants investigated here, lower tensile residual stresses were determined in contrast to room temperature processing. This is apparently due to the reduction in strength and the corresponding increase in ductility of the steel at higher temperatures. Regarding subsequent processing steps, the concave variant shows the most beneficial residual stress depth distribution over the entire depth range that could be resolved by means of the incremental hole drilling method. The convex die geometry results in largely split residual stress distribution, i.e., much higher tensile residual stresses in the axial direction in comparison with the tangential component. A clear explanation for this cannot be given based on the data presented here. However, the rather high tensile residual stress level of about 400 MPa up to a depth of approx. 200 µm stands out and should be assessed as negative in the sense of further processing steps.

### 3.2. Drawing Speed at 20 mm/s and 2000 mm/s—Material at Room Temperature

Compared to drawing at 20 mm/s at room temperature, the required drawing force is reduced at the drawing speed of 2000 mm/s due to the change in tribological conditions (Figure 8). Again, for the convex die geometry variant, the clearly higher force demand was determined in comparison to the conventional and the concave die geometry. 

In Figure 9, the microstructure of a sample drawn using convex geometry at room temperature at 2000 mm/s close to the surface is shown as an example of a section cut in longitudinal direction. At lower magnification (left), a banded structure can be seen. More meaningful, however, is the detailed view of the region very close to the surface at a significantly higher magnification (right). Here, i.e., at higher magnifications (Figure 9b), glide bands that result from the large deformations induced by the forming process can be clearly seen. In addition, locally limited deformation martensite can be seen with appropriate magnification.

In Figure 10, the fraction of austenite for the investigations of the drawing speeds is shown. Due to the higher degree of deformation, which results from the convex geometry in the near-surface area of the wire during deformation, martensite is formed when it is drawn at a speed of 20 mm/s. At the drawing speed of 2000 mm/s, a higher proportion of martensite is formed in the near-surface area, but also in both geometry variants, which is due to the much harsher forming deformations and the larger degree of deformation close to the surface, which is supported by the fact that the austenitic stainless steel exhibits a strain rate dependence of the material’s behavior.

Figure 11 shows the differences in the drawing force at speeds of 20 mm/s (left) and 2000 mm/s (right). During the continuous process section, a significantly lower drawing force can be seen at higher speeds. The influence of the forming energy on the measured drawing force is particularly significant when high elongations are achieved with high strengths, and the strengths are also strongly temperature-dependent. This applies to the austenitic chromium–nickel steel present here. The strengths decrease with increasing temperatures. Since the heat energy dissipated is time-dependent, the maximum temperatures and the temperature distribution are significantly influenced by the drawing speeds. This leads to a speed dependency of the drawing force due to the forming energy. The shorter contact times between the tool and the sample mean that more thermal energy remains in the sample.

As already explained, deformation-induced phase transformations occur with the material 1.4301 during wire drawing. This must be considered in the experimental analysis of residual stresses. These processes also depend on the drawing speed, which was taken into account in the experiments. In order to investigate this connection, residual stress analyses using the incremental hole drilling method were applied and supplemented by phase-specific XRD analysis for the near-surface region. The aim is to analyze the macro-residual stresses in a greater component depth in addition to the XRD measurements in the near-surface area. Due to the characteristic formation of martensite during cold forming of 1.4301 and the high degree of deformation at the near-surface area of the specimen due to the form elements, analyses of the retained austenite were performed by means of XRD and using the Rietveld method for data evaluation. The investigations focus on drawn wires that were produced with the geometry variants conventional (CON), convex (COV), and concave (COC), both with drawing speeds of 20 mm/s and 2000 mm/s. From a depth of approx. 30 µm (COV and COC) below the wire surface, there was already more than approx. 90% austenite by volume content. For the conventional state CON, 85% austenitic structure was already present from a depth of about 15 µm. At greater depths, the samples with an austenite content of 100% have the original structure. The retained austenite analyses on the specimen surface gave the following values: CON 81%, COV 72%, and COC 74% for the specimens drawn at 2000 mm/s. No martensite diffraction line could be determined on the surface of the samples drawn at 20 mm/s using XRD analysis. Figure 7 summarizes the residual stresses determined by X-ray analysis on the dominant austenite phase in the immediate vicinity of the surface (two material removal steps) and the results of the macro residual stress analyses using incremental hole drilling for the two main directions, axial and tangential, for the drawn wires. There are differences in the absolute values when comparing the two drawing speeds.

Typical conditions for wire drawing are 180 °C and a drawing speed of 2 m/s. These influences are examined in the paper. Previous investigations took place under laboratory conditions (room temperature 20 °C and drawing speed 20 mm/s). Now, the transfer to the serial process was examined. With reference to Figure 7 (comparison Figure 7a,b), it can be seen that as a result of an expected increase in temperature (up to 180 °C) during the transition from the laboratory process to the industrial process, a significantly lower level of tensile stress is favored. We would also like to refer to the results for Figure 11 comparison a and b. It can be seen here that at higher drawing speeds, which are typical for the wire drawing process (200 mm/s), a significant reduction in the internal tensile stress, particularly in the surface area, can be achieved with the concave drawing die geometry.

When analyzing the results, it becomes clear that wire drawing with geometry elements in the layer near the surface has significant influences on the residual stress distribution. In Figure 11, it is clearly shown that for the high drawing speeds, the axial and tangential tensile residual stresses in the near-surface region initially, i.e., close to the surface, are significantly lower and, in addition, the tangential residual stress components are much lower than the axial components. This relationship changes with increasing distance from the surface. The results clearly indicate that for the assessment of the process-induced residual stress depth distributions, residual stress data from the very surface is insufficient, not least since the values from the very-near-surface region might be strongly influenced by shear and friction effects due to the interaction between the workpiece and the tool, which strongly depends on the local and transient lubrication and, hence, friction conditions. Moreover, for the meaningful analysis of residual stresses in the near-surface regions that are affected by deformation-induced martensitic transformations of the steel 1.4301, residual stress analysis in all contributing phases (here: austenite and martensite) are mandatory to account for the multi-phase nature of the material state and to allow for calculation of the macro residual stresses for the comparison with results from continuums mechanics simulation [27,28]. On the other hand, the application of the incremental hole drilling method allows the fast determination of residual stress depth distribution down to rather large depths and provides the depth profiles of the macro residual stresses that are required for the process adjustments [29]. For the high pull rate considered in the studies, good agreement regarding the residual stress results determined by means of XRD and incremental hole drilling was achieved. When assessing the residual stress analysis carried out for the different process variants, it becomes clear that there are significant influences of the residual stress states after wire drawing, especially close to the surface. 

To evaluate the residual stress depth distributions induced by wire drawing for the wire diameter of >10 mm from the authors’ point of view, it is extremely sensible and elegant to combine phase-specific residual stress analysis by means of XRD (close to the surface) and incremental hole drilling at larger depths. The advantage of this complementary approach is that X-ray stress analysis is good for phase-specific analysis and can account for strain-induced phase transformations. In contrast, incremental hole drilling is comparably fast and yields meaningful residual stress results up to a depth of about 1–1.2 mm. For the application of X-ray stress analysis up to these depths, electrochemical layer removal causes severe redistributions of the residual stress distributions and must be taken into account. There are analytical procedures proposed in the literature to correct for this effect, e.g., following the procedure of Moore and Evans [30]. Unfortunately, they cannot be applied for the present case of a local layer removal on a cylindrical sample. That means, in most cases, in order to determine these redistributions, hard assumptions have to be made to simulate the effect, and due to this, the evaluation or correction of the effect in cylindrical samples with small diameters is often very vague.

Using this complementary analysis approach, the depth distributions shown in Figure 11 clearly show that the residual stresses strongly depend on the drawing speed and the geometry variant. In the wire drawing stage from the initial specimen diameter of D_0_ = 12 down to ø 10.8 mm and a degree of deformation of about φ = 0.26 [18] in the conventional variant, there is already a high level of tensile residual stress in the near-surface regions at a drawing speed of 2000 mm/s. The conventional geometry (CON) shows axial residual stresses of well over 650 MPa, according to the incremental hole drilling method, near the surface. By using the geometry elements with a concave additional element, this axial tensile residual stress is reduced by up to about 28% (500 MPa), especially near the region at a depth of approx. 0.1 mm; on the other hand, a peak occurs at a depth of approx. 0.3 mm. The residual stress depth distributions in samples, which were drawn with the tool staffed with a convex additional element, show a reduction in tensile residual stresses for the axial component of up to about 15% at a depth region between 0.04 and 0.32 mm. The residual stress course corresponds qualitatively to that determined for the process using the conventional geometry. The quantification of the property improvement in relation to the residual stress distribution in the wire can thus be well verified.

## 4. Conclusions

Wire drawing with specific forming elements in the dies was experimentally investigated for drawing wires of the austenitic steel 1.4301 from the initial diameter of ø 12 down to ø 10.8 (corresponds to a degree of deformation of about φ = 0.26) with two different drawing speeds. 

From the investigation, the following conclusions can be drawn:The applied complementary approach using a combination of X-ray stress analysis for residual stress analysis in the bear surface region and incremental hole drilling for a larger depth up to about 1 mm is excellently suited for fast analysis of residual stress depth distribution while taking into account the near-surface effects of deformation-induced martensitic transformations.The investigations have further shown that the restriction to the immediate surface is, in any case, insufficient to be able to evaluate the process variants sensibly.Clear differences in the residual stress distributions close to the surface occurred between the variants that were drawn using conventional, convex, and concave die geometries. Using convex geometry, a reduction in the tensile residual stresses was experimentally determined compared to the usage of conventional die geometry. This reduction can be explained in connection with the increased plastic strain when using the concave geometry element.A higher plastic strain was also reflected in the increase in the drawing force. Residual stresses can thus be achieved with this process approach already by applying individual elements in one drawing stage. Using this research approach, the properties of semi-finished products can be specifically adjusted by in-process modification of the drawing die geometries, which also implies the component properties in further manufacturing processes.The experimental investigations have further shown that the residual stresses can be characterized as an essential geometry-related feature for drawn wire samples. The residual stresses could be influenced by the specific application of individual small geometry elements in the drawing die’s forming zone.

Based on these results, a modified wire drawing process can be further developed. The sequencing and combination of several elements in several drawing stages shall be emphasized to implement the research results in a commercial wire drawing process. Only in this manner can the more favorable properties in the semi-finished wire also be reflected in the future component properties of wire products, particularly in a reduced risk of cracking at very narrow bending radii. In this context, the material model must also be expanded. The effects of the improved residual stresses have to be investigated and analyzed in specific bending experiments.

## Figures and Tables

**Figure 1 materials-17-01174-f001:**
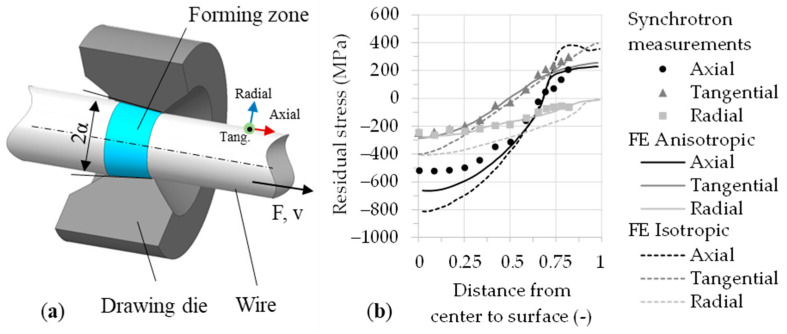
(**a**) Conventional wire drawing die (schematic view) [4]. (**b**) Typical residual stress distribution after wire drawing [5].

**Figure 2 materials-17-01174-f002:**
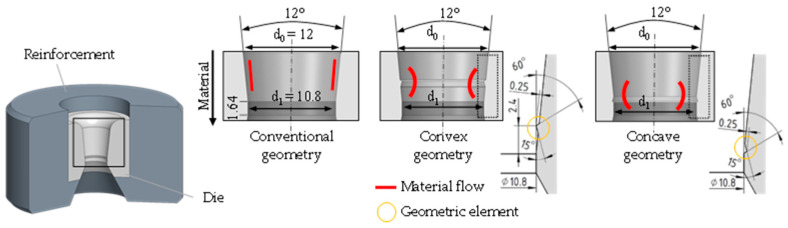
Geometry variants with special geometric elements [5].

**Figure 3 materials-17-01174-f003:**
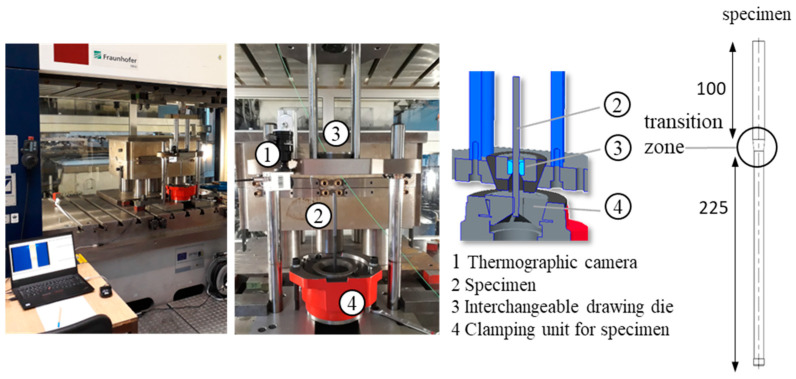
Drawing tool setup; hydraulic press; pulling speed 20 mm/s.

**Figure 4 materials-17-01174-f004:**
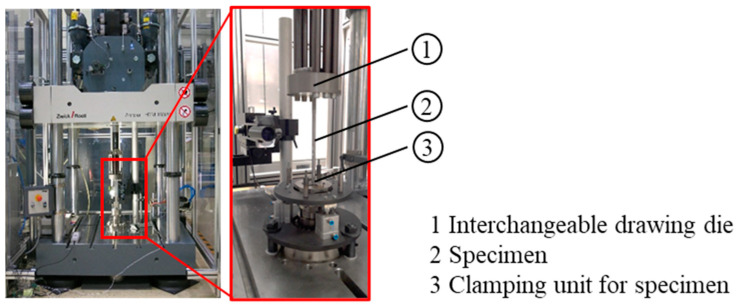
Drawing tool setup; high-speed tensile testing machine; pulling speed 2000 mm/s.

**Figure 5 materials-17-01174-f005:**
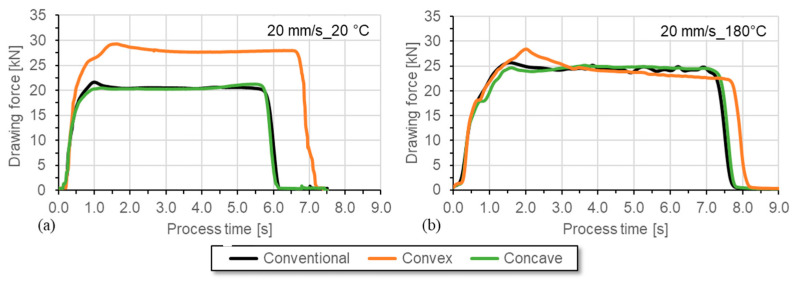
Force–process time curves for wire drawing with a drawing speed of (**a**) 20 mm/s (with lubricant) at room temperature and at (**b**) 180 °C (without lubricant).

**Figure 6 materials-17-01174-f006:**
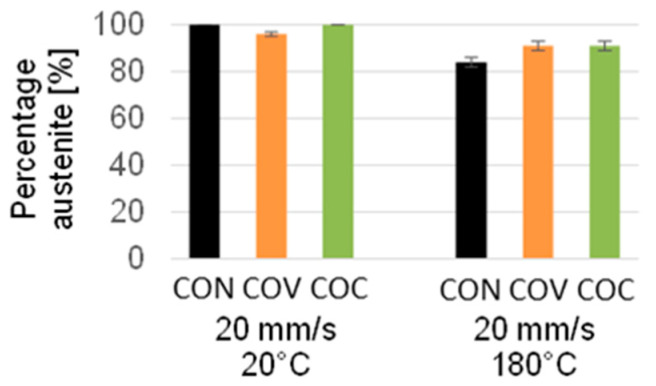
Comparison of austenite percentage depending on the wire drawing variant—for a temperature of 20 °C (room temperature) and 180 °C with drawing speed 20 mm/s (CON = conventional, COV = convex, COC = concave).

**Figure 7 materials-17-01174-f007:**
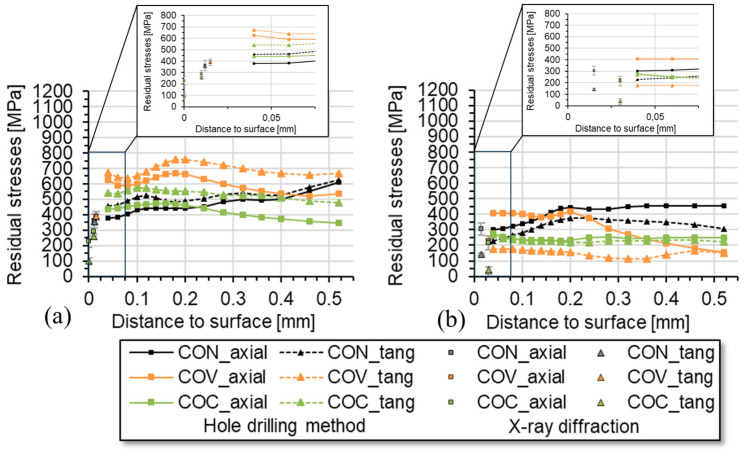
Results of XRD residual stress analysis and residual stress analyses using the incremental hole drilling method for wire drawing at (**a**) room temperature and (**b**) 180 °C (20 mm/s), (CON = conventional, COV = convex, COC = concave).

**Figure 8 materials-17-01174-f008:**
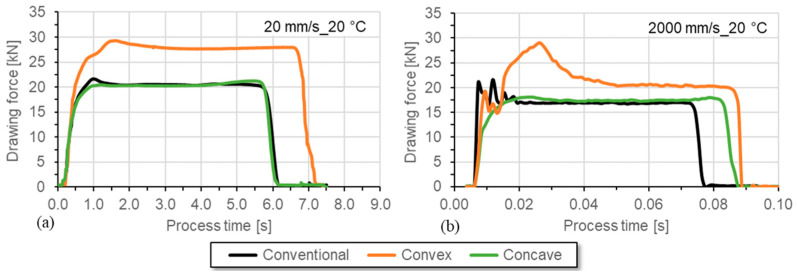
Force–process time curves for wire drawing with drawing speeds of (**a**) 20 mm/s and (**b**) 2000 mm/s at room temperature (both with same lubricant).

**Figure 9 materials-17-01174-f009:**
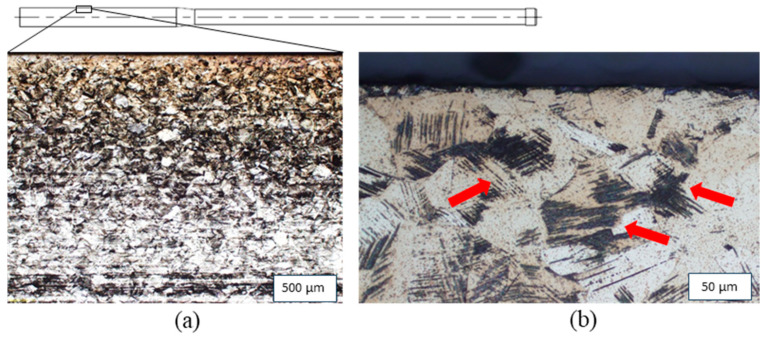
Microstructure of a sample drawn by a convex geometry: (**a**) overview image, (**b**) detailed view of the near-surface region at higher magnification. Cut in the longitudinal direction with glide bands indicated by red arrows.

**Figure 10 materials-17-01174-f010:**
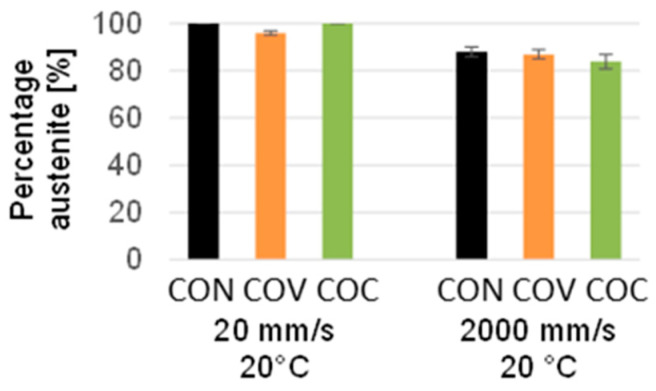
Comparison of austenite percentage depending on the wire drawing variant—at a drawing speed of 20 mm/s (**left**) and 2000 mm/s (**right**) both at room temperature (and both with the application of same lubricant), (CON = conventional, COV = convex, COC = concave).

**Figure 11 materials-17-01174-f011:**
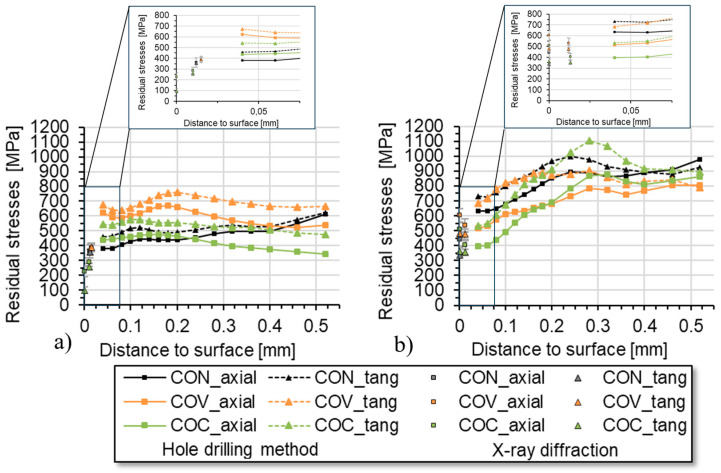
Results of XRD measurements and incremental hole drilling method for wire drawing at room temperature with drawing speed of 20 mm/s (**a**) and 2000 mm/s (**b**), (CON = conventional, COV = convex, COC = concave).

**Table 1 materials-17-01174-t001:** Chemical composition of the alloy X5CrNi18-10 (1.4301) according to DIN EN 10088-3 [20].

Element		C	Si	Mn	P	S	Cr	Ni	N
Content [wt.%]	min	-	-	-	-	-	17.5	8.0	-
	max	0.07	1.0	2.0	0.045	0.03	19.5	10.5	0.1

**Table 2 materials-17-01174-t002:** Diffraction elastic constants (DEC) used for the residual stress analysis.

Phase	hkl	s_1_(hkl)/MPa^−1^	½ s_2_(hkl)/MPa^−1^
Ferrite	{211}	−1.277 × 10^−6^	5.821 × 10^−6^
Austenite	{220}	−1.457 × 10^−6^	6.157 × 10^−6^

## Data Availability

The data presented in this study are available on request from the corresponding author. The data are not publicly available due to privacy.

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
