# Peer review of "Residual Stress Engineering for Wire Drawing of Austenitic Stainless Steel X5CrNi18-10 by Variation in Die Geometries—Effect of Drawing Speed and Process Temperature"

_materials, 2024, doi:10.3390/ma17051174_

Round 1
Reviewer 1 Report
Comments and Suggestions for Authors
Comments on the article are as follows:
Introduction
1 The introduction contains too much information about what has been done. Please make it shorter. The introduction should include the most important findings of what was determined.
Materials and Methods
1. Please justify the choice of parameters and temperatures. Why was such and not other values used?
2. Figure 3 is incomprehensible and in poor quality. Similarly, figure 4.
Results and discussion
1. why can't grease be used at an elevated temperature? Maybe there was no point in doing the tests at a higher temperature? Maybe there are special lubricants that can be used at higher temperatures.
2. why is the force requirement for convex parts slightly less compared to room temperature use?
3. why is martensitic transformation limited only to the first 10 micromeres?
4. fig.8. what is the reason for the unstable o course of forces for 2000 mm/s at the beginning of the process? Why is a clear force spike for convex observed on the graph?
5. No discussion of the obtained test results with other results, no references to the literature in particular when it comes to certain correlations that are observed in the results. These issues should be further elaborated
Conclusions
1. The conclusions should be quantified.
2. What is the industrial significance of the research results obtained? What is their significance/essence?
General comments:
- Notes in the style of In [15], (...), In [16], (...), etc. Please indicate the name(s) of the authors and show what was done. Please make changes throughout the text.
- In the text, first indicate the drawing and then show it in the text.
- Please check the figure captions as there are errors.
- Scientific articles should not use "at first sight" style wording.
- Literature/ literature review of scientific papers should include the latest scientific reports. It should include papers for 2021-2024 (i.e., the last 3 to 5 years). This makes the scientific feeling from the introduction unsatisfactory. This area needs improvement.
Reviewer 2 Report
Comments and Suggestions for Authors
The study presents the results of experimental tests conducted to analyze the influence of die geometries, drawing speed, and process temperature on the residual stress distribution in the wire drawing process. The manuscript is well structured.
Overall, the Introduction section provides a clear and concise presentation of the research problem and sets the stage for the rest of the article.
1. Here, the suggestion is about Figure 1. The circle section is better to be presented in polar coordinates. In the coordinate axes shown redial and tang directions are the same. May be tang. direction is the direction of circumferential or Hoop stress as I understand.
The authors provide detailed descriptions of the experimental setup, including the drawing tools and the conditions under which the experiments were conducted. They also explain the specific measurements and analyses used to evaluate residual stresses, such as the incremental hole drilling method and phase-specific X-ray diffraction (XRD) analysis. Additionally, the authors discuss the rationale behind their chosen methods and provide a clear justification for their experimental approach.
1. Here the position of the strain rosette is not clear. Row 216 “The measuring points were situated in the middle of the 216 drawn sample, i.e. in the region that was subjected to continuous deformation”
2. What is “differential approach” for stress calculation?
3. Rows 224-226 . There is missing Reference.
The experimental investigations include analyses of the drawing force, path, and residual stress using techniques such as incremental hole drilling and phase-specific X-ray stress analysis.
1. Here the Figures 7 and 11 are not clear. Points presenting X-ray measurements are hard to be read. May be, this part of the figure can be presented as zoomed detail.
-Measured by drill method values are very high 600-700MPa, even 1000MPa, fig11b. This is beyond the tensile strength of the material?
2. Since the methodology of the residual stress measurement and calculation is one of the main contributions I expected more detailed explanation and addition Figures with XRD difractograms etc. From figure 7 it is not clear that measurement by both methods are similar. XRD values are well lower than the drill ones.
3. Figures 6 and 10 dispersion bar is given. How much specimens or places are measured to form the confidence interval?
4. Rows 409-411 “ For the application of X-ray stress analysis up to these depths the electrochemical layer 409 removal causes severe redistributions of the residual stress distributions and must be tak-410 ing into account.” Can you site a reference for this statement?
5. Reference 29 is not mentioned in the text
The findings from these experiments are used to draw conclusions regarding the effectiveness of modifying the wire drawing process to tailor residual stress distributions in the drawn wires, but the I think that written in the bullet is not supported by the results.
“
• Clear differences in the residual stress distributions close to the surface occurred be-443 tween the variants that were drawn using conventional, convex and concave die ge-444 ometries. Using the convex geometry, a reduction of the tensile residual stresses was 445 experimentally determined compared to the usage of a conventional die geometry. 446 This reduction can be explained in connection with the increased plastic strain when 447 using the convex geometry element. “
My overall impression is that the text and some of the figures should be improved. Calculated values for the residual stresses should be checked again.
Author Response
"Please see the attachment."

Round 2
Reviewer 1 Report
Comments and Suggestions for Authors
I would like to thank the authors for their responses. They are satisfactory, so in my opinion the article can be published in its present form.
Author Response
Thank you for your assessment that in your opinion the article can be published in its current form.
Attached you will find the Review Round 2, where the comments of the academic editor's have been taken into account.
